# Merits of hiPSC-Derived Cardiomyocytes for In Vitro Research and Testing Drug Toxicity

**DOI:** 10.3390/biomedicines10112764

**Published:** 2022-10-31

**Authors:** Ping-Hsien Wang, Yi-Hsien Fang, Yen-Wen Liu, Min-Long Yeh

**Affiliations:** 1Department of Biomedical Engineering, National Cheng Kung University, Tainan 70140, Taiwan; 2Division of Cardiology, Department of Internal Medicine, National Cheng Kung University Hospital, College of Medicine, National Cheng Kung University, Tainan 70403, Taiwan; 3Center of Cell therapy, National Cheng Kung University Hospital, College of Medicine, National Cheng Kung University, Tainan 70403, Taiwan; 4Institute of Clinical Medicine, College of Medicine, National Cheng Kung University, Tainan 70403, Taiwan

**Keywords:** cardiotoxicity, human-induced pluripotent stem cell (hiPSC), cardiomyocyte (CM), cardiac differentiation, H9c2, hepatocellular carcinoma (HCC)

## Abstract

The progress of medical technology and scientific advances in the field of anticancer treatment have increased the survival probabilities and duration of life of patients. However, cancer-therapy-induced cardiac dysfunction remains a clinically salient problem. Effective anticancer therapies may eventually induce cardiomyopathy. To date, several studies have focused on the mechanisms underlying cancer-treatment-related cardiotoxicity. Cardiomyocyte cell lines with no contractile physiological characteristics cannot adequately model “true” human cardiomyocytes. However, applying “true” human cardiomyocytes for research is fraught with many obstacles (e.g., invasiveness of the procedure), and there is a proliferative limitation for rodent primary cultures. Human-induced pluripotent stem-cell-differentiated cardiomyocytes (hiPSC-CMs), which can be produced efficiently, are viable candidates for mimicking human cardiomyocytes in vitro. We successfully performed cardiac differentiation of human iPSCs to obtain hiPSC-CMs. These hiPSC-CMs can be used to investigate the pathophysiological basis and molecular mechanism of cancer-treatment-related cardiotoxicity and to develop novel strategies to prevent and rescue such cardiotoxicity. We propose that hiPSC-CMs can be used as an in vitro drug screening platform to study targeted cancer-therapy-related cardiotoxicity.

## 1. Introduction

Cancer and cardiovascular diseases are the leading causes of morbidity and mortality worldwide [1]. Owing to the huge advances in anticancer treatments, the prognosis of cancer patients and their survival duration are dramatically improving. Therefore, the importance and clinical significance of cancer-therapy-induced cardiac dysfunction cannot be overstated.

Understanding the mechanisms of cancer-therapy-related cardiotoxicity is essential to prevent cancer-therapy-related cardiotoxicity and provide optimal care for cancer survivors with cardiac dysfunction. Human cardiomyocytes are recognized as a powerful model for in vitro physiological and pathophysiological studies; however, many obstacles lie ahead before “true” human cardiomyocytes can be applied for research. First, obtaining primary human cardiac tissues requires invasive procedures such as myocardial biopsy or open-heart surgery. Therefore, there are major ethical concerns regarding obtaining human cardiomyocytes for research. Second, it is difficult to culture primary human cardiomyocytes in vitro. Even though the human cardiomyocyte cell line is commercially available, it is not recommended to expand the cultured human cardiomyocytes or to culture them for a long time since these cells are not proliferative in vitro. The difficulties in obtaining cardiac tissue from patients and the inability to propagate human cardiomyocytes in vitro limit the utility of human cardiac tissue and/or human cardiomyocytes for investigating the mechanisms of cardiac disease.

Currently, there are some cardiomyocyte cell lines available, such as atrial cardiomyocyte cell lines (including HL-1 and AT-1 cells derived from rodent atrial tumors), rat ventricular cardiac myocytes (H9c2 cells), and human cardiomyocytes [2,3]. Notably, mouse atrial cells and rat ventricular cardiac myocytes are immortalized proliferative cells, and these cells do not spontaneously contract or fire action potential [3,4]. As a result, these cardiac myocyte cell lines are not reflective of “true” human cardiomyocytes.

Human embryonic stem cells obtained from fertilized eggs are well-known to be capable of differentiating into cardiomyocytes; however, their application for research purposes is constrained by social and ethical considerations. Somatic cells, a favorable alternative for research, reprogrammed into induced pluripotent stem cells (iPSCs) by transfecting four transcription factors (Sox2, Oct3/4, Klf4, and c-Myc), have the ability to differentiate into three germ layers with each type of organ tissue [5]. Recently, iPSCs have been applied in various medication study fields, including tissue engineering of cell regeneration [6,7], disease models [8,9,10], and drug screening [11,12,13,14,15]. However, drug screening studies have focused on toxicity or clinical side effects, not on the pharmacological mechanism [16,17,18,19]. Human-induced pluripotent-stem-cell (hiPSC)-derived cardiomyocytes (hiPSC-CMs) possess characteristics similar to those of primary human cardiomyocytes [20,21]. Thus, hiPSC-CMs may be good candidates for investigating the pathophysiology and molecular mechanisms of many acquired cardiac diseases, such as cancer-treatment-related cardiotoxicity. We conducted this study to demonstrate the merit of hiPSC-CMs as a platform for in vitro mechanistic studies and drug toxicity tests.

## 2. Materials and Methods

### 2.1. Differentiation of hiPSC-CMs from Human PSCs

hiPSCs in a feeder-free system were obtained from the Human Disease iPSC Service Consortium, Academia Sinica, Taiwan. The hiPSCs grew in Matrigel (Corning^®^catalog no. 354230; Corning, NY, USA)-coated cell culture dishes with Stemflex medium (Thermo Fisher Scientific; catalog no. A3349401; Waltham, MA, USA) subsequently incubated at 37 °C in a 5%-CO_2_ atmosphere. We used Accutase (Innovative Cell Technologies; catalog no. AT-104; San Diego, CA, USA) to detach those cells that reached high confluence and supplemented these detached cells with fresh medium to enable continuous culture propagation. For suspended hiPSCs attached to the culture dish, the ROCK inhibitor Y27632 (Sigma-Aldrich; catalog no. Y0503; St. Louis, Mo, USA) was added to the culture medium.

hiPSC-CMs were subjected to a well-established Wnt signaling modulation protocol using a directed differentiation method [22,23]. Briefly, hiPSCs were cultured in RPMI medium (Thermo Fisher Scientific; catalog no. 22400089) with the Wnt agonist CHIR 99021 (5 μM; Stemcell Technology; catalog no. 72054) in the early stages of differentiation, followed by the Wnt antagonist IWR-1 (5 μM; Sigma-Aldrich; catalog no. I0161) to induce cardiomyocyte differentiation. During cell differentiation, serum-free RPMI-B27 (minus insulin; Thermo Fisher Scientific; Catalog no. 17504044) was used. After 14 days of in vitro differentiation, cells were dispersed using 0.05% trypsin–EDTA and reseeded. Cultures were fed every other day thereafter with serum-free RPMI-B27 (insulin plus B27; Thermo Fisher Scientific; catalog no. A1895601). Only cell preparations containing ≥80% cardiac-troponin-T (cTnT)-positive cardiomyocytes (determined by flow cytometry) were used in subsequent studies.

H9c2 cells were purchased from the Bioresource Collection and Research Center (BCRC) and cultured in Dulbecco’s modified Eagle’s medium (DMEM, Thermo Fisher Scientific; catalog no. 11965092) with 10% fetal bovine serum (FBS, Thermo Fisher Scientific; catalog No. A5256701), 100 U/mL penicillin, and 100 μg/mL streptomycin (Life Technologies Corporation; catalog no. 10378016) at 37 °C in an atmosphere of 95% air and 5% CO_2_.

### 2.2. Sorafenib and Doxorubicin Solution Preparation

Sorafenib (Bayer Healthcare Pharmaceutics Inc., Whippany, NJ, USA) and doxorubicin (Cayman Chemical; catalog No. 15007, Ann Arbor, Michigan, USA) were prepared according to the manufacturer’s instructions. Both sorafenib and doxorubicin solutions were prepared as 10 mM stocks in a glass vial. The stock solution was mixed vigorously for 10 min at room temperature. For testing, the pharmaceutical compounds were diluted in a glass vial using an external solution (137 mM NaCl, 4 mM KCl, 1 mM MgCl_2_, 10 mM glucose, 1.8 mM CaCl_2_, and 10 mM HEPES). Dilutions were prepared within 30 min of treatment. Equal amounts of DMSO (0.1%) were used as vehicle controls.

### 2.3. Cell Viability Assay

Approximately 10,000 hiPSC-CMs or H9c2 cells were seeded in Matrigel-precoated 96-well culture plates with 100 μL of RPMI-B27 supplement at 37 °C for two days. Cells were cultured with 0~100 μM sorafenib and doxorubicin in dose- and time-dependent manners. Colorimetry was used to study the cell viability. According to the manufacturer’s instructions, we added 20 μL of tetrazolium compound solution [3-(4,5-dimethylthiazol-2-yl)-5-(3-carboxymethoxyphenyl)-2-(4-sulfophenyl)-2H-tetrazolium, inner salt; MTS] (Progema; catalog no. G3582, Madison, WI, USA) and 5% PMS solution (Sigma-Aldrich; catalog no. P9625) into the culture medium at 37 °C for 2 h. The optical density at 490 nm was measured using a continuous wavelength analyzer (Marshall Scientific; SpectraMax; catalog no. 34PC384, Hampton, NH, USA).

### 2.4. Flow Cytometry

The dissociated cells were resuspended and washed by phosphate-buffered saline (PBS). The cells were fixed in 4% paraformaldehyde (PFA; Sigma-Aldrich; catalog no. 158127) for 10 min on ice. After washing twice with PBS, 0.1% Triton X-100 was used to permeabilize the cells. Fluorescent-conjugated antibodies, anti-cTnT (BD; catalog no.565618, Franklin Lakes, NJ, USA), and the ventricular isoform of myosin regulatory light-chain 2 (MLC2v; BD; catalog no. 565497) were used for staining. All positive fluorescent signals were detected and quantified using a BD Canto II flow cytometry system.

### 2.5. Immunofluorescent Staining

Human iPSC-CMs were fixed in 4% paraformaldehyde for 20 min, followed by washing (PBS) wash. Nonspecific antigenic sites of the fixed cells were blocked with 5% FBS (FBS; Gibco, Gaithersburg, MD, USA) for 1 h at 4 °C. Primary antibodies, including alpha-actinin (Abcam; catalog no. ab68194, Cambridge, UK) and connexin 43 (Thermo Fisher Scientific; catalog no. 71-0700) were hybridized at 4 °C overnight. The samples were then rinsed with PBS and incubated with secondary antibodies: goat anti-mouse IgG (H&L), Alexa Fluor™ 488 (Abcam; catalog no. ab175473), and goat anti-rabbit IgG (H&L), Alexa Fluor 568 (Abcam; catalog no. ab175471), at room temperature for 1 h. All primary and secondary antibodies were prepared in PBS containing 2% BSA (Thermo Fisher Scientific, catalog no. 15260037). Fluorescence images were captured using an inverted fluorescence microscope (OLYMPUS; catalog no. BX51; Tokyo, Japan). The acquired fluorescent images were processed and quantified using ImageJ software.

### 2.6. hiPSC-CMs Contractility Measurement

A camera was mounted on an inverted microscope with slow-motion features (120 frames/s) to record a video of the beating hiPSC-CMs for offline analysis. All the acquired videos were analyzed using the automated nonprofit software MUSCLEMOTION [24]. We used MUSCLEMOTION software v1.0 to read and convert the videos into uncompressed AVI files and measured the shrinkage curve to obtain the contraction amplitude and velocity of hiPSC-CMs.

### 2.7. Seahorse Assay

The mitochondrial respiratory function was assessed according to the manufacturer’s instructions. Briefly, hiPSC-CMs were plated at a density of 2 × 10^4^ cells per well on a Matrigel-coated seahorse assay energy meter plate with bicarbonate-free RPMI, and H9c2 cells were plated at a density of 2 × 10^4^ cells per well in DMEM with 10% FBS culture medium.

Seahorse XF-24 (Seahorse Bioscience; Santa Clara, CA, USA) was used to measure aerobic respiration and acidification rates. First, we seeded 2 × 10^4^ hiPSC-CMs on a Matrigel-coated seahorse assay energy meter plate with bicarbonate-free RPMI. These hiPSC-CMs were washed twice and incubated in the assay medium for 1 h prior to the experiment. The detection probe was activated at 37 °C overnight in a dark CO_2_-free tank. Oligomycin (Sigma-Aldrich; catalog no. O4876) and carbonyl cyanide 4-(trifluoromethoxy) phenylhydrazone (FCCP; Sigma-Aldrich; catalog no. C2920) were used to assess mitochondrial-dependent aerobic respiration and to calculate the ATP production rate. Antimycin (Sigma-Aldrich; catalog no. A8674) was used as a baseline correction.

### 2.8. Reactive Oxygen Species (ROS) Quantification

hiPSC-CMs were plated at a density of 2 × 10^4^ cells per well on a Matrigel-coated 96- plate with bicarbonate-free RPMI culture medium at 37 °C in an atmosphere of 95% air and 5% CO2 overnight. ROS generation was detected using the ROS Detection Cell-Based Assay Kit: 2′,7′Dichlorodihydrofluorescein diacetate (DCFDA) (Thermo Fisher Scientific, ab113851). A final concentration of 20 μM DCFDA was added to the culture medium after the hiPSC-CMs were treated with sorafenib or doxorubicin. Plates were incubated in the dark for 45 min at 37 °C in an atmosphere of 95% air and 5% CO2. Dichlorodihydrofluorescein levels were assessed using an ex/Em = 485/535 nm fluorescence plate reader (Fluoroskan Ascent, Thermo Fisher Scientific). H_2_O_2_ (100 Mm) was used as the positive control for ROS generation.

### 2.9. Electrophysiological Measurements

On the day of the electrophysiological experiments, hiPSC-CMs and H9c2 cells were harvested and transferred to a homemade recording chamber mounted on the stage of a CKX-41 inverted microscope (Olympus; Yuanyu, Taipei, Taiwan). The cells were bathed in normal Tyrode’s solution at room temperature (22–25 °C). Standard patch clamp experiments were performed using an RK-400 patch amplifier (Bio-Logic, Claix, France). The recording electrodes were fabricated from Kimax-51 borosilicate capillaries (#3450; Kimble; Dogger, Taipei, Taiwan) using a PP-830 vertical puller (Narishige; Taiwan Instrument Cp., Taipei, Taiwan), and were then fire-polished with an MF-83 microforge (Narishige; Taiwan Instrument Cp., Taipei, Taiwan). A tip resistance ranging between 3 and 5 MΩ was chosen for recording.

### 2.10. Electrophysiological Data Recordings and Analysis

The current and potential signals were monitored using a digital oscilloscope (model 1602; Pchome eBay Co., Ltd., Taipei, Taiwan). Data were stored on a laptop at 10 kHz using an acquisition interface (Digidata-1440A; Molecular Devices, Sunnyvale, CA, USA) and subsequently analyzed using Pclamp 10.7 (Molecular Devices). The computer was placed on top of an adjustable Cookskin stand (Ningbo, Zhejiang, China) for efficient operation during recording. The current signals were low-pass-filtered at a frequency of 3 kHz. The data digitally acquired during each experiment were later analyzed using different analytical tools, including the LabChart 7.0 program (AD Instruments; KYS Technology, Tainan, Taiwan). To normalize for cell size, the currents (Pa) were reported as the current densities (Pa/Pf).

### 2.11. Western Blotting

Sorafenib-treated hiPSC-CMs and H9c2 were lysed with 0.1 M Tris, 0.3 M NaCl, 0.1% SDS, 0.5% sodium deoxycholate, and 1% Triton X-100 in a cocktail of antiproteases (Sigma-Aldrich Corporation, St. Louis, MO, USA). Lysates were incubated for 5 min at 100 °C and centrifuged by 15,000× *g* for 10 min. Proteins were quantified using the Pierce™ BCA Protein Assay Kit (Thermo, catalog no. 23225), and 30 μg of total protein was separated on a 10% sodium dodecyl sulfate–polyacrylamide gel. The proteins were transferred to polyvinylidene difluoride (PVDF) membranes and blocked with 5% skim milk at room temperature for 1 h. The PVDF membrane was incubated with the following primary antibodies: cleaved caspase 3 (Cell Signaling; catalog no. 9661, Danvers MA, USA) and caspase 9 (Cell Signaling; catalog no. 20750), LC3B (Cell Signaling; catalog no. 2775), and GAPDH (Cell Signaling; catalog no. 2118) overnight at 4 °C. HRP-conjugated anti-rabbit (Abcam, catalog no. ab97051) and anti-mouse (Abcam, catalog no. ab205719) secondary antibodies were incubated at room temperature for 1 h. It was visualized using an enhanced chemiluminescence substrate (Thermo Fisher Scientific; catalog no.32209). Images were acquired using a ChemiDoc XRS (Kodak, Rochester, NY, USA). Protein was quantified using ImageJ software (Bethesda, MD, USA).

### 2.12. Statistics

In vitro experiments were performed in triplicate. For quantitative comparisons, data were presented as the mean ± standard error of the mean and compared using one-way analysis of variance. Multiple comparisons between groups were determined using the Bonferroni *t*-test by unpaired experimental design, parametric test, and two-tailed calculations. Differences were considered statistically significant at *p* < 0.05.

## 3. Results

### 3.1. Generation and Characterization of hiPSC-CMs

hiPSCs were expanded using Matrigel-coated culture plates with stemflex medium. When the cultured cells reached confluence, we used a serum-free, monolayer direct cardiac differentiation protocol involving serial application of B27 minus insulin with the Wnt agonist CHIR 99021 in the early stages of differentiation, followed by the Wnt antagonist IWR-1 (Figure 1a) [25] for cardiomyocyte differentiation. On day 21 of cardiac differentiation, the differentiated cells were harvested for characterization. A comparison of hiPSC-CM and H9c2 cell morphology is shown (Figure 1b). To identify the characteristics of these cells, flow cytometry was used to analyze cardiac-specific markers (cTnT and MLC2v). Data showed that only hiPSC-CMs expressed both cardiac-specific markers, but not H9c2 cells. (Figure 1c). α-actinin influences the mechanics of the cytoskeleton by cross-linking actin filaments and other cytoskeleton components expressed in the hearts. Connexin 43 (Cx43) is essential for heart structure formation. However, only hiPSC-CMs detected both markers by immunofluorescence staining (Figure 1d), which indicated the presence of possible electrical communication between hiPSC-CMs. Huh7 was established from a well-differentiated hepatocyte-derived cellular carcinoma cell line that is often used as a model to investigate hepatocellular carcinoma. We chose the Huh7 cell line for the sorafenib dosage control test. The IC50 of sorafenib in Huh7 cells was approximately 5–6 μM, but H9c2 cells and hiPSC-CMs had a higher risk acceptance (2–3-fold) of sorafenib dosage compared to Huh7 (Figure 1e). Treatment with 4 μM sorafenib in Huh7 induced cell death within 24 h (Figure 1f), but no effect was observed in H9c2 cells or hiPSC-CMs (Figure 1f,g). However, when these cells, including Huh7 cells, H9c2 cells, and hiPSC-CMs, were treated with doxorubicin, there was no significant difference in cell viability among these three different cells (Appendix A).

### 3.2. Contractile Function Is Impaired in Sorafenib-Treated and Doxorubicin-Treated hiPSC-CMs

Since the observation that 4 μM sorafenib induced Huh7 cell death, hiPSC-CMs compared with H9c2 cells have become our research motivation for drug testing platforms to determine whether both cells are affected.

On day 14 of cardiac differentiation, we acquired images of beating hiPSC-CMs for analysis, using the same approach as that for H9c2 cells. Automated nonprofit software, MUSCLEMOTION, was used to quantify the contractility and relaxation [24]. There was a substantial disparity of contraction amplitude, contraction velocity and relaxation velocity between hiPSC-CMs and H9c2 cells (Figure 2). Furthermore, H9c2 cells and hiPSC-CMs were treated with doxorubicin. Only hiPSC-CMs demonstrated a significant decline in contractile and relaxation function (Appendix A). Nevertheless, H9c2 cells do not possess functional beating characteristics. There was no difference in contractile and relaxation function between those H9c2 cells treated and not treated with sorafenib or doxorubicin (Figure 2 and Appendix A). From a physiological perspective, H9c2 cells are not ideal for drug toxicity screening. (Sorafenib treatment caused a decrease in hiPSC-CM contractility, as shown in the supplemental data.)

### 3.3. Bioenergetics of Mitochondrial Function in hiPSC-Derived Cardiomyocytes and H9c2 Cells

Mitochondria constitute almost 95% of the cellular energy supply in the form of ATP. ATP production or consumption in hiPSC-derived cardiomyocytes was determined by simultaneous measurement of dissolved oxygen in the media, expressed as the oxygen consumption rate (OCR). Figure 3a shows the results for the OCR in dose-dependent sorafenib culture for 24 h. Clinically, the effective dose of sorafenib therapy is 400 mg/day, with a relative quantity of approximately 4–6 μM in vitro at the cellular level. Sorafenib as a target drug affects only cancer cells, but our data from the seahorse assay showed that over 4 μM sorafenib in the treatment of hiPSC-CMs caused ATP consumption and reflexed on basal respiration (Figure 3b), which corresponds to the result of ATP-linked respiration (Figure 3d). hiPSC-CMs were insusceptible to nonmitochondrial respiration (nonmitochondrial oxygen consumption) following high-dose sorafenib treatment, which clarified the effect of sorafenib-induced toxicity on mitochondrial damage (Figure 3c). Maximal respiration represents the extreme operational effectiveness of mitochondria, showing no significance in sorafenib-treated hiPSC-CMs by dose (Figure 3e). Proton migration into the matrix without producing ATP renders the coupling of substrate oxygen and ATP generation incomplete. Unbalanced ATP production led to aberrant proton leakage in sorafenib-treated hiPSC-CMs (Figure 3f). Despite the productive measurement of the OCR in H9c2 cells (Figure 3g), physiological parameters may not be ideal. No step of the bioenergetics of mitochondrial function was significantly different between those H9c2 cells undergoing sorafenib treatment and those cells not undergoing sorafenib treatment (Figure 3h–l).

### 3.4. ROS Generation in Sorafenib- or Doxorubicin-Treated hiPSC-CMs

Intracellular ROS detection in cells was based on the oxidation of nonpolar-cell-permeable DCFDA. The hiPSC-CMs were treated with 4 μM sorafenib overnight or 0.5 μM doxorubicin for 6 h. Using a DCFDA assay, we detected ROS production in the sorafenib-treated hiPSC-CMs and the doxorubicin-treated hiPSC-CMs (Figure 4 and Appendix A).

### 3.5. The Electrophysiological (EP) Cardiac Characteristics Present in hiPSC-CMs but Not H9c2 Cells

To minimize intracellular milieu, cell-attached current recordings were performed. The potential was maintained at the level of the resting potential (approximately −70 mV). Before the electrophysiological experiments, hiPSC-CMs were treated with 4 μM sorafenib for 24 h. The frequency of spontaneous action currents (ACs) recorded in sorafenib-treated hiPSC-CMs was significantly lower than that in sorafenib-untreated hiPSC-CMs. (AC firing frequency: sorafenib-treated hiPSC-CMs vs. non-sorafenib-treated hiPSC-CMs, 0.11 ± 0.02 Hz vs. 0.35 ± 0.04 Hz, *p* < 0.05, Figure 5a,b).

### 3.6. In Contrast to the H9c2 Cells, Using hiPSC-CMs for Drug Toxicity Screening Is a Closer Milestone in Cell Molecular-Level Research

LC3s (MAP1-LC3A, B, and C) are widely used as biomarkers of autophagy, in which relevant autophagosomes exist in different locations, and LC3B proteins are equally distributed throughout the cytoplasm and localized in the nucleolar regions [26]. LC3 has been shown to have mutual effects on RIPK1 and RIPK3 proteins in the myocardium and cardiomyocytes that cause necroptosis activation. Under hypoxia, LC3 induces hypoxic myocardial injury and other hypoxia-related diseases [27]. In our results, caspase9 was expressed in a dose- (Figure 6a,b) and time-dependent manner (Figure 6c,d). Theoretically, hiPSC-CMs and H9c2 cells, which are normal cells, are not affected by sorafenib-induced apoptosis; however, after sorafenib treatment with a dose exceeding 10 μM (Figure 6a) for 24 h or 4 μM sorafenib treatment for 48 h (Figure 6c), cleaved caspase 9 expression was detected. In comparison, H9c2 cells were not sensitive to sorafenib-induced toxicity, except for treatment with 4 μM sorafenib for 72 h. Cleaved caspase 3 was further detected by sorafenib-induced apoptosis in hiPSC-CMs over for does over 8 μM (Figure 6a) for 72 h of treatment (Figure 6c), but not in H9c2 cells. Damaged mitochondria release cytochrome c (Cyt c), one of the prominent factors in the apoptotic scene, which contributes to the apoptotic dismantling of the cell, was observed. Cyt c mediates the apoptosis-protease-activating factor 1 necessary for proteolytic maturation of caspase-9 and caspase-3 [28]. The synthesized caspases in cells as inactive zymogens undergo proteolytic cleavage to be fully activated. Caspase-9 and caspase-3 are the initiator and downstream effector caspases involved in intrinsic apoptosis, respectively. Therefore, we speculated that caspase-3, activated by caspase-9, entails a single proteolytic cleavage between the large and small subunits of caspase-3 [29]. Caspase 9 activates disassembly in response to insults, triggering the emancipation of cytochrome c from the mitochondria [30] (Figure 6e).

## 4. Discussion

We demonstrated the advantages of hiPSC-CMs as a platform for in vitro research and drug toxicity tests. There are many choices of cardiac cells for in vitro studies, either from cell lines or from primary cultures. The largest proportion of cardiotoxicity studies use the H9c2 cell line, which belongs to the cardiac myoblast, a primary culture from rat ventricular tissue. Another choice of cardiomyocyte-like cells is HL-1, an immortalized mouse cardiomyocyte cell line that retains the phenotype of cardiac cells and serves as a useful model for examining cardiac pathophysiology. However, these cells originate from the atrium rather than the ventricle. Commercialized human cardiomyocyte cell lines (AC16) or primary cultures from rodents are available for cardiac-related studies. Although it may be easy to obtain these cells for examination, a crucial flaw is that these cells do not recapitulate the physiological significance of cardiac ventricle contraction. In contrast, hiPSC-CMs possess many of the same characteristics as those of primary human cardiomyocytes (Table 1). Thus, hiPSC-CMs could be good candidates for investigating the pathophysiology and molecular mechanisms of many acquired cardiac diseases, such as cancer-treatment-related cardiotoxicity.

Targeted therapy drugs can be grouped according to how they work or which part of the cell they target. A rough classification list is as follows. (a) Signal transduction inhibitors: this class of inhibitors blocks the molecular activities that participate in signal transduction. Some malignant cancer cells are stimulated to divide continuously, without being prompted to do so by external growth factors. The process by which a cell responds to environmental signals interferes with inappropriate signaling. (b) Angiogenesis inhibitors: tumors grow beyond a certain size, relying on the blood supply. Oxygen and nutrients from the blood vessels provide tumors with continued growth. Angiogenesis inhibitors can block the growth of new blood vessels and impede tumor growth. (c) Apoptosis-inducing drugs: apoptosis is a natural process in the body that eliminates unnecessary or abnormal cells, but cancer cells have strategies to avoid apoptosis. The use of apoptosis-inducing drugs to control cancer cell death is another approach. (d) Immunotherapy drugs: some immunotherapy drugs are monoclonal antibodies that can recognize specific molecules on the surface of cancer cells. Once a monoclonal antibody binds to the target molecule, the cells expressing the target molecule are destroyed. Some other monoclonal antibodies that bind to certain immune cells may help these immune-combined cells kill cancer cells. (e) Monoclonal antibodies attached to toxins: monoclonal antibodies specifically deliver toxic molecules that cause cancer cell death. Because the antibody binds to its target cell, the toxic molecule linked to the antibody ultimately kills that cell. This occurs regardless of the strategies used to prevent, restrict, or kill cancer cells, normal cells, or cells that undergo drug treatment in the human physiological microenvironment. Sorafenib is known to induce cardiac toxicity [31]. In addition, sorafenib combined with other anticancer drugs, such as lovastatin, can produce synergistic cytostatic/cytotoxic effects against tested tumor cell lines [32]. The combination of sorafenib and another anticancer drug, imatinib, impaired mitochondrial function in isolated rat cardiac myofibers and H9c2 cells. Damage to enzyme complexes in the mitochondrial electron transfer system may be due to the accumulation of toxicants in the inner mitochondrial membrane, causing mitochondrial ROS production, mitochondrial proteins, and dysfunction, eventually leading to apoptosis in H9c2 cells [33]. Sorafenib treatment alone was effective in reducing disease progression in the early stages of treatment of thyroid cancer, but cardiac monitoring was recommended [34]. In a follow-up anticancer drug-induced cell toxicity study in vitro, sorafenib uptake was associated with a decrease in cytosolic calcium concentrations, SR calcium loading, and phospholamban S16 phosphorylation, indicating the direct cardiomyocyte-intrinsic toxicity of sorafenib in the human myocardium [35].

## 5. Limitations

Because AC16 cells and hiPSC-CMs have human genetic backgrounds, it may be appropriate to compare the effect of anticancer therapy on these two cells. However, we did not have AC16 cells at hand, so we did not carry out such a comparison. We recognize that this is a significant limitation of this study.

## 6. Conclusions

hiPSC-CMs can be used to investigate the pathophysiology and molecular mechanisms of cancer-treatment-related cardiotoxicity and to develop novel strategies to prevent and rescue anticancer-therapy-related cardiotoxicity. Our findings demonstrate that hiPSC-CMs are suitable for drug screening and may be used as an in vitro research platform for mechanistic studies (Figure 7).

## Figures and Tables

**Figure 1 biomedicines-10-02764-f001:**
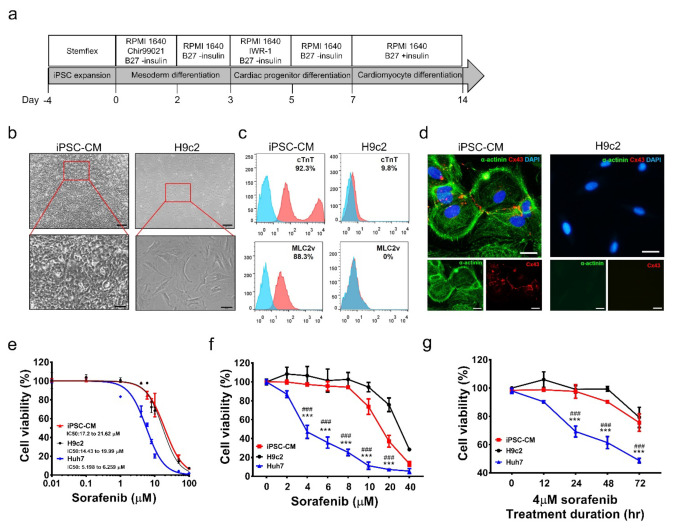
Comparison of the characteristics of hiPSC−differentiated cardiomyocytes and H9c2 cells with drug-induced toxicity by sorafenib in vitro. (**a**) Schematic of the iPSC−CM differentiation protocol. (**b**) Establishment and validation of hiPSC and differentiated cardiomyocytes compared with H9c2 cells. The morphology shows the difference between iPSC−CMs and H9c2 cells. (**c**) Marker confirmation of iPSC−CMs and cardiac myoblast (H9c2 cells) of cardiac active protein Troponin T (cTnT) and ventricular marker Myosin light-chain 2 (MLC2v) by flow cytometry. Blue wave shows isotype control and red shows cardiomyocyte marker. (**d**) Representative fluorescent images of cardiomyocyte markers: alpha-actinin (α-actinin, green), connexin 43 (Cx43, red), and nucleus staining, DAPI (blue) of hiPSC−differentiated cardiomyocyte and H9c2 cells, scale bar: 50 μM. (**e**) IC50 of sorafenib treatment in iPSC−CMs, H9c2 cells, and Huh7 after 72 h. (**f**) Sorafenib-treated dose-dependent cell viability after sorafenib treatment for 72 h. (**g**) Sorafenib-treated time-dependent cell viability. Each data point represents the mean ± SEM (*n* = 3), *** *p* < 0.001 compared with iPSC-CMs, ### *p* < 0.001 compared with H9c2 cells.

**Figure 2 biomedicines-10-02764-f002:**
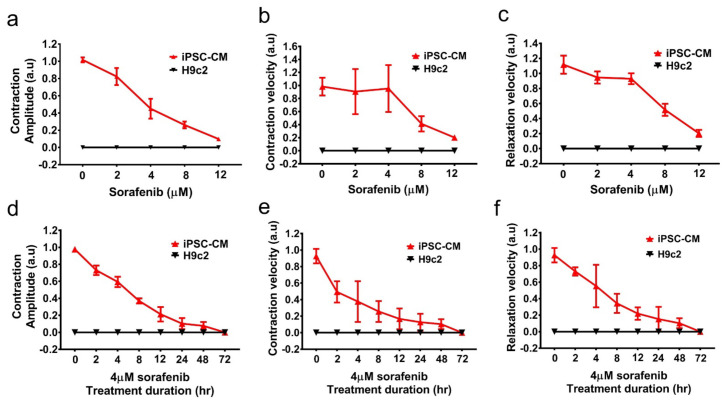
Contractile dysfunction in sorafenib-treated iPSC−CMs. (**a**–**c**) Dose-dependent muscle motion of iPSC−CMs and H9c2 cells under 0, 2, 4, 8, and 12 μM sorafenib treatment for 24 h. (**a**) Contraction amplitude, (**b**) contraction velocity, and (c) relaxation velocity. (**d**–**f**) Time-dependent muscle motion of iPSC−CMs and H9c2 cells treated with 4 μM sorafenib for 72 h. (**d**) Contraction amplitude, (**e**) contraction velocity, and (**f**) relaxation velocity. Each data point represents the mean ± SEM (*n* = 3).

**Figure 3 biomedicines-10-02764-f003:**
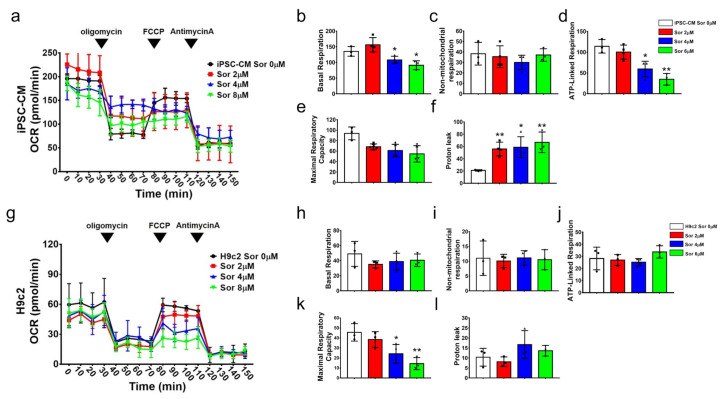
Bioenergetic parameters in hiPSC-derived cardiomyocytes and H9c2 cells. iPSC-CMs (**a**–**f**) and H9c2 cells (**g**–**l**) under 0, 2, 4, and 8 μM sorafenib treatment for 24 h. Idealized bioenergetic profiling trace demonstrating oxygen consumption rate (OCR) (**a**,**g**), basal respiration (**b**,**h**), nonmitochondrial oxygen consumption(**c**,**i**), ATP production (**d**,**j**), Maximal Respiratory Capacity (**e**,**k**), and proton leak (**f,l**). Each data points represent the mean ± SEM (*n* = 3). *p*-values: * *p* < 0.05, ** *p* < 0.01 compared with 0 μM sorafenib treatment.

**Figure 4 biomedicines-10-02764-f004:**
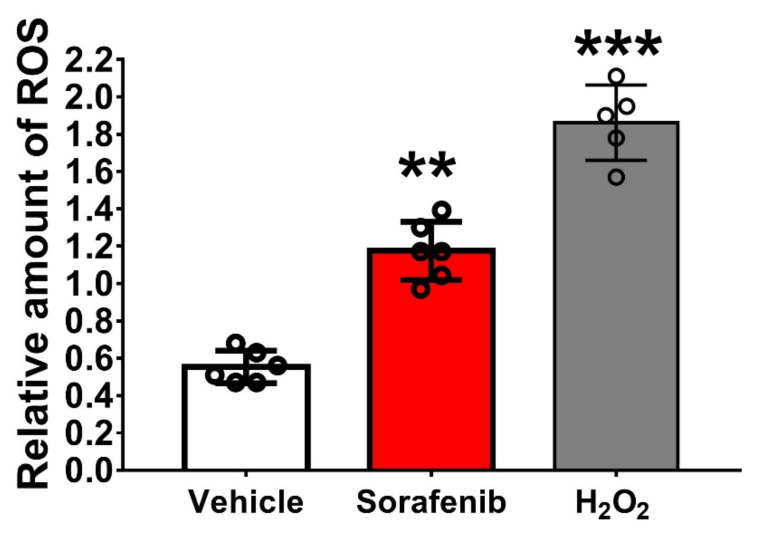
The quantification of ROS generated in sorafenib-treated hiPSC-CMs. Mitochondria-damage-induced reactive oxygen species (ROS) production can be detected in both sorafenib-treated hiPSC-CMs. The imbalance between mitochondrial reactive oxygen species overexpression and removal leads to oxidative damage affecting cellular components. Since the cardiomyocyte is a highly energetic organ, it is vulnerable to damage caused by oxidative stress. *p*-values, ** *p* < 0.01, *** *p* < 0.001, compared with vehicle.

**Figure 5 biomedicines-10-02764-f005:**
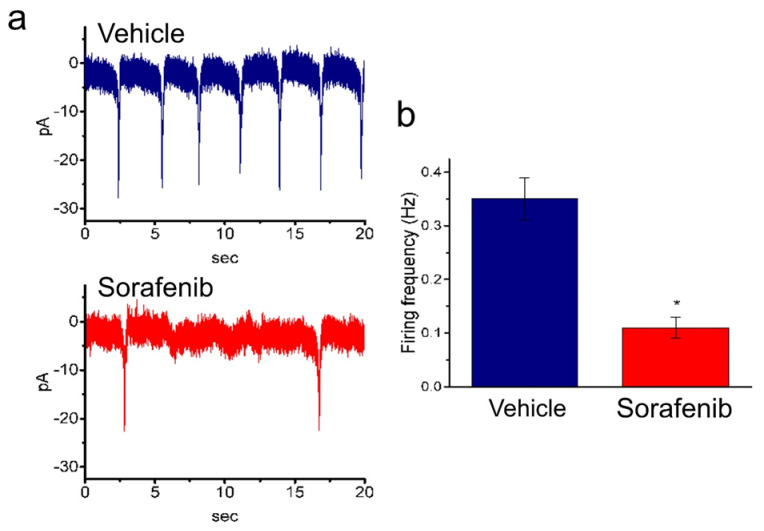
Action current (AC) firing recording in human-induced pluripotent-stem-cell-derived cardiomyocytes (hiPSC−CMs) with or without treatment of sorafenib. Prior to the electrophysiological experiments, hiPSC−CMs were incubated in 4 μM sorafenib for 24 h. This set of experiments were conducted in cell-attached current recordings, and the potential was held at the level of the resting potential (approximately −70 mV). (**a**) Representative current traces obtained in untreated (**a**) and treated (**b**) hiPSC−CMs. The downward deflection shows the occurrence of AC. (**b**) Summary bar graph showing the firing frequency of untreated and untreated cells (mean ± SEM; *n* = 7 for each bar). * Significantly different from untreated cells (*p* < 0.05).

**Figure 6 biomedicines-10-02764-f006:**
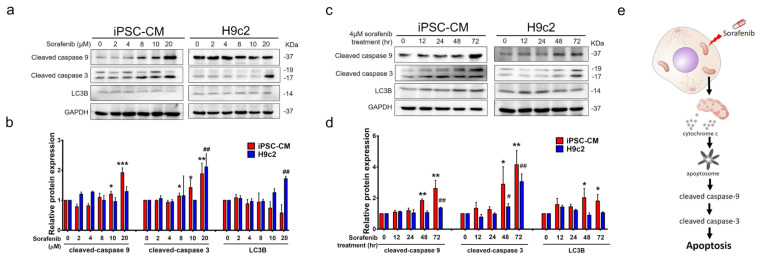
Apoptosis markers of cleaved caspase 9, cleaved caspase 3, and autophagy marker LC3B expressed in hiPSC−CMs and H9c2 cells dose- and time-dependently. (**a**) hiPSC−CMs and H9c2 cells treated with sorafenib under 2, 4, 8, 10, and 20 μM for 72 h. Apoptosis markers (cleaved caspase 9, cleaved caspase 3) and autophagy marker (LC3B) were analyzed by Western blot. (**b**) Quantification of cleaved caspase 9, cleaved caspase 3, and LC3B expression from diagram a. *p*-values, * *p* < 0.05, ** *p* < 0.01, *** *p* < 0.001, compared with 0 μM sorafenib in hiPSC−CMs., ## *p* < 0.01, compared with 0 μM sorafenib in H9c2 cells. Data are shown as the mean ± SEM. (**c**) hiPSC−CMs and H9c2 cells treated with sorafenib under 4 μM for 0,12,24,48, and 72 h. Apoptosis markers (cleaved caspase 9, cleaved caspase 3) and autophagy marker (LC3B) were analyzed by Western blot. (**d**) Quantification of cleaved caspase 9, cleaved caspase 3, and LC3B expression from diagram c. *p*-values, * *p* < 0.05, ** *p* < 0.01, 4 μM sorafenib compared with 0 μM sorafenib in hiPSC−CMs. # *p* < 0.05, ## *p* < 0.01, 4 μM sorafenib compared with 0 μM sorafenib in H9c2 cells. Data are shown as the mean ± S.E.M. (**e**) Schematic representation of the postulated mechanism of action of sorafenib.

**Figure 7 biomedicines-10-02764-f007:**
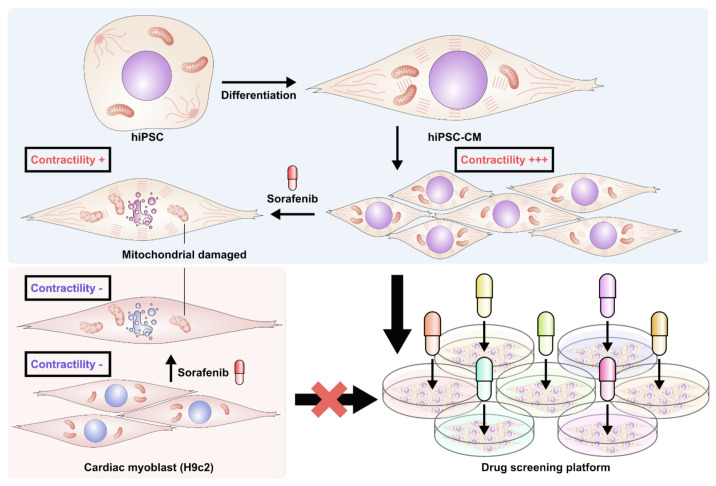
Diagrammatic illustration of preponderance of drug screening platform by iPSC-induced cardiomyocytes.

**Table 1 biomedicines-10-02764-t001:** The comparison of different sources of cardiac cells.

Cell Type	iPSC-CM	H9c2	Adult Cardiomyocyte
Morphology	resemble of fetal CMs	myoblast	bundle structure
Development level	Immature cardiomyocyte	myoblast	Mature cardiomyocyte
Contractility	Yes	No	Yes
Cardiac markers expression	Yes	No	Yes
Action potential	Yes	No	Yes
ATP source	glucose/fatty acid	glucose	fatty acid
Metabolism	Anaerobic/aerobic respiration	Anaerobic respiration	aerobic respiration

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
