# Peer review of "Merits of hiPSC-Derived Cardiomyocytes for In Vitro Research and Testing Drug Toxicity"

_biomedicines, 2022, doi:10.3390/biomedicines10112764_

Round 1

Reviewer 1 Report

In the present study, authors tried to demonstrate the merit of iPSC derived cardiomyocytes in determing drug toxicity. 

1. The authors claimed a drug screening platform of iPSC derived cardiomyocytes, however, only one drug sorafenib was investigated.  Thus, other drugs or drug candidates should be included to support the current conclusions.

2. It is necessary to include electrophysiological assay to further determine the drug toxicity in iPSC derived cardiomyocytes.

3. In the part of mitochondrial function, ROS generation should be considered in iPSC derived cardiomyocytes with or without drug treatment.

4. Some abbreviations need to show full name upon their first appearence, for example "PSC".

5. "hiPSC in feeder free system obtained from Human Disease iPSC Service Consortium, Academia Sinica, Taiwan" should be  "hiPSC in feeder free system obtained from Human Disease iPSC Service Consortium, Academia Sinica, Taiwan, China"

Author Response

Responses to reviewer 1 comments:

  1. The authors claimed a drug screening platform of iPSC derived cardiomyocytes, however, only one drug sorafenib was investigated. Thus, other drugs or drug candidates should be included to support the current conclusions.

Response: We appreciated your important comment. We provided the data of hiPSC-derived cardiomyocytes treated with doxorubicin in the manuscript and the supplementary data (lines 277-280, 304-309 and supplementary Figures 1 and 2,).

  1. It is necessary to include electrophysiological assay to further determine the drug toxicity in iPSC derived cardiomyocytes.

Response: Thank you for your point. We added electrophysiological assay data in the revised version (Figure 5, lines 370-377).

  1. In the part of mitochondrial function, ROS generation should be considered in iPSC derived cardiomyocytes with or without drug treatment.

Response: Thank you for your comment. We have checked mitochondrial function and ROS generation as your suggestion. The data are shown in Figure 4 and Supplementary Figure 3 (Lines 355-357).

  1. Some abbreviations need to show full name upon their first appearance, for example "PSC".

Response: I apologized for our fault. English editing of our manuscript has been done.

  1. "hiPSC in feeder free system obtained from Human Disease iPSC Service Consortium, Academia Sinica, Taiwan" should be "hiPSC in feeder free system obtained from Human Disease iPSC Service Consortium, Academia Sinica, Taiwan, China".

Response: Thanks for your comment. After checking the official website of Academia Sinica, its official name is “Academia Sinica, Taiwan, Republic of China" (https://www.sinica.edu.tw/en/articles/12). We have revised it in our manuscript.

Reviewer 2 Report

The manuscript “The merit of hiPSCs derived cardiomyocytes for in vitro research and drug toxicity test” shows that hiPSC derived cardiomyocytes can be used as a platform for invitro drug testing and toxicity analysis. The authors have provided experimental evidence towards their claim using the drug sorafenib. However, the manuscript should undergo major revision and following are a few comments which would help in improving it.

Major comments-

1.      English needs to be checked. For example in the abstract, the sentence “ Since medical technology and the progress of science technology of anti-cancer treatments, patients survive opportunities and duration of lives increased.” is not clear. I think the authors meant “ The progress in medical technology and anti-cancer treatments has led to better patient survival and increased life-expectancy.” It is extremely difficult to follow the paper because of incorrect usage of English language.

2.      Authors should provide a review of current literature on hiPSC derived cardiomyocytes in the introduction regarding the various protocols to make them, their applications such as in drug testing etc.

3.      The authors should have compared the effect of sorafenib on a human cardiomyocyte cell line like AC16 in addition to rat cardiomyocyte cell line H9c2. This will be a proper comparison given that both hiPSC derived CM and AC16 have human genetic background.

Minor comments-

1.      All quantifications like cell viability, contraction amplitude should include the DMSO control results and representative images for the DMSO control should also be shown.

Author Response

Responses to Reviewer 2 Comments and Suggestions for Authors

  1. English needs to be checked. For example in the abstract, the sentence “ Since medical technology and the progress of science technology of anti-cancer treatments, patients survive opportunities and duration of lives increased.” is not clear. I think the authors meant “ The progress in medical technology and anti-cancer treatments has led to better patient survival and increased life-expectancy.” It is extremely difficult to follow the paper because of incorrect usage of English language..

Response: I am sorry to make you being confused. To improve our manuscript’s readability, English editing has been done. English editing certificate has been listed (FYI).

  1. Authors should provide a review of current literature on hiPSC derived cardiomyocytes in the introduction regarding the various protocols to make them, their applications such as in drug testing etc.

Response: Thank you for your important point. We have added a relative paragraph in the revised introduction (page 2, line 65-79).

  1. The authors should have compared the effect of sorafenib on a human cardiomyocyte cell line like AC16 in addition to rat cardiomyocyte cell line H9c2. This will be a proper comparison given that both hiPSC derived CM and AC16 have human genetic background. 

Response: We appreciate your important recommendation. We do agree that it will be a proper comparison between hiPSC-derived cardiomyocytes and AC16 cells. However, because we do not have AC16 cell line in our institute and we are unable to get AC16 cells immediately, thus we could not do such a comparison right now. Therefore, we have mentioned this in the “Discussion”  session (lines 488-492).

Minor comments

All quantifications like cell viability, contraction amplitude should include the DMSO control results and representative images for the DMSO control should also be shown.

Response: We do agree your comment that all quantifications should include the “DMSO” control results. In our study, the vehicle groups contained 0.1% DMSO which has been described in the “Sorafenib solution preparation” method of the “Materials and Methods” session. (line 137)

Round 2

Reviewer 1 Report

The authors have addressed my concerns properly. The current manuscript is acceptable for publication.

Reviewer 2 Report

The authors have answered most of the concerns raised.